# The Association between Infant Colic and the Multi-Omic Composition of Human Milk

**DOI:** 10.3390/biom13030559

**Published:** 2023-03-18

**Authors:** Desirae Chandran, Kaitlyn Warren, Daniel McKeone, Steven D. Hicks

**Affiliations:** Department of Pediatrics, Penn State College of Medicine, Hershey, PA 17033, USA

**Keywords:** *Lactobacillus*, colic, breastmilk, microbiome, cytokines, microRNA, inflammation

## Abstract

Infant colic is a common condition with unclear biologic underpinnings and limited treatment options. We hypothesized that complex molecular networks within human milk (i.e., microbes, micro-ribonucleic acids (miRNAs), cytokines) would contribute to colic risk, while controlling for medical, social, and nutritional variables. This hypothesis was tested in a cohort of 182 breastfed infants, assessed with a modified Infant Colic Scale at 1 month. RNA sequencing was used to interrogate microbial and miRNA features. Luminex assays were used to measure growth factors and cytokines. Milk from mothers of infants with colic (*n* = 28) displayed higher levels of *Staphylococcus* (adj. *p* = 0.038, *d* = 0.30), miR-224-3p (adj. *p* = 0.023, *d* = 0.33), miR-125b-5p (adj. *p* = 0.028, *d* = 0.29), let-7a-5p (adj. *p* = 0.028, *d* = 0.27), and miR-205-5p (adj. *p* = 0.029, *d* = 0.26) compared to milk from non-colic mother–infant dyads (*n* = 154). Colic symptom severity was directly associated with milk hepatocyte growth factor levels (*R* = 0.21, *p* = 0.025). A regression model involving let-7a-5p, miR-29a-3p, and *Lactobacillus* accurately modeled colic risk (*X*^2^ = 16.7, *p* = 0.001). Molecular factors within human milk may impact colic risk, and provide support for a dysbiotic/inflammatory model of colic pathophysiology.

## 1. Introduction

Infantile colic is classified as a functional gastrointestinal disorder with recurrent symptomatic episodes that lack identifiable organic cause. The 2016 Rome IV criteria define colic as a behavioral syndrome involving persistent episodes of crying, fussiness, and irritability in otherwise healthy infants that resolves by five months of age [1]. Prevalence of colic has been found to vary, ranging from 5% to 20% [2,3]. Stress resulting from colic places a significant burden on infants, their families, and the healthcare system [4]. Inability to soothe the crying infant may increase caregiver frustration, straining relationships within the family unit; in severe cases, this can lead to physical abuse toward the infant [5]. 

Since the etiology of infantile colic is currently unknown, the 2015 American Family Physician guidelines for its clinical diagnosis recommend initially excluding other organic conditions before utilizing subjective measures of caregiver symptom reports [6]. Even when a diagnosis can be made, there are no evidence-based treatments that fully resolve symptoms. Instead, the current treatment guidelines emphasize counseling caregivers on the self-limited nature of colic and mitigating infant symptoms until they cease [6,7,8]. However, these methods of symptom mitigation also vary in efficacy, as they are based upon theories of causation that do not wholly explain the colic outcome [9,10,11]. One novel treatment involves administering the probiotic *Lactobacillus reuteri* DSM17938, which has been implicated in assuaging imbalances within the developing intestinal microbiota; however, there is conflicting evidence as to its therapeutic potential for breastfed infants, and little evidence demonstrating benefits for bottle-fed infants [12]. Hydrolyzed formula has been found to significantly decrease the length of crying episodes in bottle-fed infants, but the difficult distinction between colic and milk protein allergy obscures this association [13,14]. The lack of objective diagnostic criteria and evidence-based treatment strategies for infantile colic reflects the need for greater understanding of its pathophysiologic origins, which is vital to establish targeted therapies and improve outcomes. 

There is some evidence that the composition of human milk may modulate infant gastrointestinal symptoms, and impact infant temperament [15,16]. Preliminary studies indicate that modifications in maternal diet may improve infant colic by altering carbohydrate composition in human milk [17]. However, it is likely that this relationship is driven by a complex interplay between the microbiome and metabolic substrates, which may, in turn, drive host inflammatory responses [18,19]. Intestinal dysbiosis has been described at the species and phylum level in infants with dysbiosis [18]. A randomized control trial of perinatal probiotic administration in 66 breastfeeding mothers found reduced levels of interleukin 6, interleukin 10, and transforming growth factor beta 1 (TFGβ1) in human milk, and reduced risk of colic symptoms in breastfeeding infants [20].

There is growing recognition of human milk as a complex biologic system with immunomodulatory properties [21,22]. Aside from commonly recognized inflammatory proteins (e.g., cytokines), another class of molecules that are ubiquitous in milk and play a vital role in immunomodulation, are micro-ribonucleic acids (miRNAs) [23,24]. Protected within extracellular vesicles, miRNAs in human milk have the ability to enter the infant gastrointestinal (GI) tract and impact gene expression [25,26]. In fact, abnormal fluctuations in the expression of specific miRNAs have been consistently correlated with both presence and severity of GI disorders [27]. However, to our knowledge, there are no studies that have investigated the relationship between human milk miRNA levels and infant colic. Further, no study has employed a multi-omic approach to determine how microbial, cytokine, and miRNA components within human milk may interact to modulate infant health. 

The goal of this investigation was to identify novel human milk components that modulate risk for infantile colic. We hypothesized that the levels of specific miRNAs, microbes, and cytokines within human milk would differ between breastfeeding mothers of infants with colic and breastfeeding mothers of infants without colic. Further, we posited that these multi-omic factors would modulate colic symptoms, while controlling for medical/demographic factors, social determinants of health, and other nutritional components (e.g., maternal diet, formula introduction) (Figure 1). To test this hypothesis we performed a prospective cohort study involving 182 breastfeeding mother–infant dyads.

## 2. Materials and Methods

### 2.1. Study Design

This study involved a secondary analysis of data collected as part of the Breast Milk IMPACT (Influence of the Micro-transcriptome Profile on Atopy in Children over Time) Study [28]. The Breast Milk IMPACT study involved a convenience sample of 221 breastfeeding infants enrolled within 7 days of delivery and followed until 12 months of age. Eligibility criteria were term delivery (37–42 weeks gestation) and maternal intention to breastfeed beyond four months. Exclusion criteria were maternal conditions that could impact breastfeeding (e.g., drug addiction, HIV), infant conditions that could impact breastfeeding (e.g., cleft lip/palate, metabolic disease, NICU admission >7 days), factors affecting long-term follow up (e.g., plan for primary pediatric care outside our medical center, or infant adoption), and inability to complete surveys (i.e., non-English speaking). 

### 2.2. Enrollment

Enrollment occurred between 20 April 2018 and 10 May 2020 at the newborn nursery or the outpatient pediatrics clinics affiliated with our academic medical center. Ethical approval was obtained from the Penn State Institutional Review Board (STUDY00008657). Participating mothers provided written, informed consent. There were 2487 infants screened for eligibility, 359 who met criteria, 221 who consented to participate, and 182 who completed the study. Study completion was defined as contribution of a milk sample, completion of medical and demographic surveys, and verification of the primary medical outcome. The primary medical outcome for this secondary analysis was presence or absence of infant colic at one month, defined by clinical evaluation by a board-certified clinician using standard (Rome IV) criteria, and confirmation through parent report on an abbreviated version of the Infant Colic Scale [29]. Clinician diagnosis of infant colic was extracted from the electronic medical record by research staff (Figure 2).

### 2.3. Data Collection

Medical and demographic characteristics were collected from mother–infant dyads using electronic surveys administered by research staff at enrollment. The following medical and demographic characteristics were collected: maternal age, infant biological sex, birth weight (g), family history of food allergy, and delivery mode (vaginal or cesarean section). Weight-for-length Z-score on the World Health Organization growth curve was extracted from the electronic medical record and change from birth to one month of age was determined in order to assess adequacy of milk supply (which could impact infant temperament). Character and severity of colic symptoms was assessed at one month using an abbreviated version of the Infant Colic Scale [29], containing two sub-scales: “Immature Gastrointestinal System” and “Difficult Infant Temperament”. These sub-scales were specifically chosen to reduce survey burden, while allowing assessment of both infant temperament and co-morbid gastrointestinal symptoms that can mimic colic (i.e., constipation and gastroesophageal reflux). The two sub-scales contained a total of 12 items, measured on a six-point Likert scale, from which a modified total score was derived. The National Survey of Lead and Allergens in Housing (NSLAH) was used to assess the following social determinants of health that could impact reports of infant temperament: maternal education, maternal marital status, infant health insurance status, infant racial/ethnic minority status, household income, and number of persons living in the household [30].

### 2.4. Milk Collection

Human milk samples were collected approximately 1 month post-delivery. Milk (1–5 mL) was manually expressed from a sterilized nipple surface (with soap and water) into RNAse-free tubes prior to feeding (i.e., fore-milk), as we have previously described [31]. Previous investigations have found minimal differences in miRNA content between fore- and hind-milk; however, we exclusively utilized pre-feed (hind-milk) samples to minimize confounding. Samples were immediately transferred to −20 °C, underwent 1 freeze-thaw cycle for aliquoting, and were placed at −80 °C while awaiting molecular analysis. 

### 2.5. Cytokine Processing

Samples were spun for 20 min at 4 °C at 200 rcf to separate lipid, skim, and cellular fractions. Skim milk fractions were spun further at 16,000 g for eight minutes to ensure removal of any lipid or cellular fractions prior to analysis. The skim milk fraction was used to measure six cytokines, selected based upon their bioavailability in human milk, and their potential relevance to inflammatory conditions in the infant gut [32,33,34,35]. An automated immunoassay (ProteinSimple, San Jose, CA, USA) was used to measure levels of chemokine C-C Motif Ligand 5 (CCL5; RANTES), hepatocyte growth factor (HGF), and interleukin-8 (IL-8) per manufacturer instructions. We used a commercially available, custom 12-plex human magnetic bead-based multiplex assay (Bio-Techne, Minneapolis, MN) to measure interleukin 4 (IL-4), transforming growth factor beta 1 (TGF-β1), and transforming growth factor beta 2 (TGF-β2). Following standard sample activation procedure using HCl and NaOH/Hepes, a multi-species performance magnetic bead-based multiplex assay was employed per manufacturer instructions. All measurements were acquired on a Luminex MAGPIX instrument running the xPONENT software, version 4.3.

### 2.6. Microbial and microRNA Processing

For each sample, 50 μL of the lipid fraction was used for RNA extraction. We focused on the lipid fraction of MBM based upon findings from our lab and others, demonstrating that this fraction contains robust concentrations of miRNAs with high potential for maternal–infant transfer [25,26,31]. RNA purification was performed using a Norgen Circulating and Exosomal RNA Purification Kit (Norgen Biotech; Thorold, ON, Canada), as previously described [36]. RNA quality was confirmed on an Agilent Bioanalyzer 2100 (Agilent, Santa Clara, CA, USA). RNA was sequenced at the SUNY Molecular Analysis Core using the Illumina TruSeq Small RNA Prep protocol and a NextSeq500 instrument (Illumina; San Diego, CA, USA) at a targeted depth of ten million, 50 base, single-end reads per sample. Reads were aligned to the hg38 build of the human genome using Partek Flow (Partek; St. Louis, MO, USA) and the Bowtie2 aligner. RNAseq was selected to permit interrogation of all known miRNAs within the human genome, and allow interrogation of microbial RNAs using the remaining un-aligned RNAs. Mature miRNA counts within each sample were quantified with miRBase v22. Microbial RNAs within each sample were quantified with K-Slam, using the NCBI Taxonomy database at the genus level. Read quality score and read quantity were used to confirm quality control prior to analysis. The miRNA and microbial features with consistent detection (raw read counts ≥ 10 in ≥10% of samples) were quantile normalized and mean-center scaled. Down-stream analysis focused on six miRNAs (Let-7a-5p, miR-125b-5p, miR-224-3p, miR-205-5p, miR-29a-3p, miR-199a-3p) [37,38,39,40,41], and six microbial features (Shannon Alpha Index, *Mycoplasma*, *Lactobacillus*, *Staphylococcus*, *Clostridium*, *Escheria*) with potential relevance to infant colic based on previous studies identifying their importance in nociception or inflammatory/irritable bowel conditions [42,43,44,45].

### 2.7. Statistical Analysis 

Medical/demographic characteristics, social determinants of health, and nutritional factors were compared between infants with colic and their peers using a Student’s *t*-test, a Mann–Whitney test, or a chi-square test, as appropriate. A Mann–Whitney test was used to compare milk levels of miRNAs, cytokines, and microbes between groups. False discovery rate correction was used to adjust p-values for multiple testing. Molecular factors were also assessed for associations with symptom severity on the modified Infant Colic Scale using Spearman’s Rank Correlation testing. Feed-backward logistic regression was used to assess the ability of multi-omic factors to predict the presence or absence of infant colic. Criteria for inclusion in the final model were *X*^2^ > 4.0 and *p* < 0.05. Factors were assessed for collinearity. Discriminative accuracy of the model was visualized on a receiver operator characteristic curve. Area under the curve (AUC), sensitivity, and specificity were reported.

## 3. Results

### 3.1. Participants

Participating infants were predominantly female (107/182, 58%), born via vaginal delivery (146/182, 80%), with an average birth weight of 3357 (±569) grams (Table 1). Average maternal age was 29 (±4) years. Most infants gained adequate weight in the first month after delivery (mean change in weight-for-length Z-score was 1.12 ± 1.5), and few had a family history of food allergy (17/182, 9%). Nearly one-quarter of participants self-identified as a racial or ethnic minority (43/182, 23%), and few reported an income below the federal poverty level (10/170, 5.8%). Most mothers were married (149/182, 81%), with a college diploma (131/182, 72%), and private health insurance (157/182, 86%). The median household size was four persons (range: 2–9). Nearly one-third of mothers introduced some formula within one month of delivery (57/182. 31%). Infants with colic (11/28, 39%) were more likely to identify as a racial/ethnic minority than infants without colic (32/154, 20%; *p* = 0.034, *X*^2^ = 4.50). The two groups did not differ in any other medical/demographic factors, social determinants of health, or nutritional characteristics.

Infants with colic had a higher average score on the modified Infant Colic Scale (43 ± 13) than infants without colic (31 ± 9; *p* = 2.9 × 10^−7^). Mothers of infants with colic were more likely to agree or strongly agree with the statement, “When my baby starts to fuss, nothing I do helps” (*p* = 4.6 × 10^−6^). However, there was no difference between colic and non-colic groups in reports of “My baby vomits undigested milk” (*p* = 0.07) or “My baby has no difficulty passing stool” (*p* = 0.15). These results indicate that the colic group was physiologically representative of colic, rather than gastroesophageal reflux, or constipation (which are often confused with colic). 

### 3.2. Multi-Omic Characteristics of Human Milk from Mothers of Infants with Colic

#### 3.2.1. Milk Collection Variables

To ensure that circadian rhythm and milk maturity did not confound molecular characteristics of human milk from mothers of infants with colic, we recorded hour of milk collection and infant age at milk collection. There was no difference between the time of day at which milk collection occurred for infants with colic and infants without colic (*p* = 0.51). There was no difference in infant age at the time of collection (*p* = 0.67).

#### 3.2.2. Proteomics

There was no difference in the levels of CCL5/RANTES (*p* = 0.080, *d* = 0.23), HGF (*p* = 0.067, *d* = 0.24), IL8 (*p* = 0.19, *d* = 0.17), TGFβ1 (*p* = 0.76, *d* = 0.05), TGFβ2 (*p* = 0.29, *d* = 0.18), or IL4 (*p* = 0.80, *d* = 0.04) in the milk from mothers of infants with colic versus those without colic. Only human milk levels of HGF were directly associated with the modified infant colic score (*R* = 0.21, *p* = 0.025; Figure 3).

#### 3.2.3. Microbiome

There was no difference in microbial diversity in the milk from mothers of infants with colic versus those without colic (*p* = 0.46, *d* = 0.08). Levels of *Staphylococcus* were higher in the milk from mothers of infants with colic, compared to mothers of infants without colic (*p* = 0.0076, adj. *p* = 0.038, *d* = 0.30; Figure 4). There was no difference in the levels of *Mycoplasma* (*p* = 0.21. *d* = 0.14), *Lactobacillus* (*p* = 0.23, *d* = 0.14), *Clostridium* (*p* = 0.35, *d* = 0.10), or *Escherichia* (*p* = 0.36, *d* = 0.10) between the two groups. No microbial factors were associated with severity of colic symptoms on the infantile colic score (*p* > 0.05). 

#### 3.2.4. Human Milk miRNAs

Levels of four miRNAs in human milk differed between mothers of infants with colic and mothers of infants without colic. Levels of miR-224-3p (*p* = 0.0038, adj. *p* = 0.023, *d* = 0.33), miR-125b-5p (*p* = 0.010, adj. *p* = 0.028, *d* = 0.29), let-7a-5p (*p* = 0.018, adj. *p* = 0.028, *d* = 0.27), and miR-205-5p (*p* = 0.019, adj. *p* = 0.029, *d* = 0.26) were all higher in the milk from mothers of infants with colic (Figure 5). Levels of miR-29a-3p (*p* = 0.19, *d* = 0.13) and miR-199a-3p (*p* = 0.87, *d* = 0.05) did not differ between groups, and no miRNA levels correlated with severity of colic symptoms on the modified infant colic score (*p* > 0.05).

### 3.3. Multi-Omic Modeling of Colic Risk

A feed-backward regression approach was used to identify the multi-omic factors implicated in infant colic. A binomial model employing one microbe and two miRNAs displayed a significant association with presence or absence of colic (*X*^2^ = 16.7, AIC = 143, BIC = 156, *p* = 0.001). Elevated milk levels of miR-29a-3p (*Z* = 2.25, *X*^2^ = 4.8, *p* = 0.027) and let-7a-5p (*Z* = 3.25, *X*^2^ = 11.6, *p* = 0.001) were associated with a greater likelihood of colic, whereas elevated levels of *Lactobacillus* in human milk reduced the likelihood of infant colic (*Z* = 1.78, *X*^2^ = 6.1, *p* = 0.013). The three factors predicted colic outcomes with 76% accuracy (79% sensitivity and 59% specificity).

## 4. Discussion

This study harnessed a complex array of medical/demographic factors, social determinants of health, nutritional components, and molecular profiles of human milk to identify novel biologic interactions that may play a role in infant colic. We identified milk miRNAs, microbes, and cytokines that may contribute to colic risk in infants predisposed by social factors, such as racial/ethnic minority status. Specifically, elevated milk levels of HGF were associated with higher levels of colic symptoms. Higher levels of miR-29a-3p and let-7a-5p were associated with increased risk of colic development, whereas elevated levels of *Lactobacillus* appeared to confer a protective effect. The impact of these multi-omic factors on colic outcomes exceeded many medical/demographic and social factors previously implicated in colic risk. 

Although this is the first study to collectively assess the relationship between human milk microbes, miRNAs, and cytokines with infant colic, our findings are largely consistent with prior studies interrogating these individual molecular features to nociception and gut inflammation. For example, a randomized placebo-controlled trial of 80 infants found that oral administration of *Lactobacillus reuteri* DSM 17938 resulted in a 2.5-fold relative risk reduction in infant colic at 28 days [46]. Similarly, in the present study, levels of *Lactobacillus* were associated with a 1.78-fold reduction in colic risk. 

These results also showed that elevated human milk levels of miR-29a-3p increased risk for infant colic. These findings are consistent with a study of human intestinal epithelium from patients with diarrhea-predominant irritable bowel syndrome, demonstrating increased levels of miR-29a-3p were associated with down-regulation of tight junction protein ZO-1 and claudin-1, which could be reversed through miR-29a-3p inhibition [47]. Thus, elevated miR-29a-3p in human milk may lead to “leaky gut” pathophysiology, which has been implicated in a number of inflammatory GI conditions [48]. 

There is emerging evidence that human milk HGF plays an important role in establishment of the infant GI tract [49]. A study of fetal small intestinal cells cultured with human milk found that levels of HGF were associated with a significant trophic epithelial response, which exceeded the effects of recombinant HGF alone. In our study, human milk HGF levels were associated with the level of infant colic symptoms. This relationship may represent a compensatory response of human milk composition to unique infant needs, a phenomenon, which has been well described for both nutritional and non-nutritional milk components [50,51,52]. For example, a study of term, breastfeeding infants found that levels of hypoxanthine, xanthine, and thiocyanate in infant saliva interact with human milk to generate reactive oxygen species and nucleosides capable of regulating microbes in the infant gastrointestinal tract [52]. Intriguingly, microbial targets included both *Staphylococcus aureus* and commensal *Lactobacillus*.

A commonly proposed mechanism for infant colic involves dysbiosis, inflammation, and heightened pain sensitivity [53,54]. Indeed, the current study identifies bacterial pathogens (i.e., *Staphylococcus*), and inflammatory miRNAs (i.e., let-7a-5p) which are elevated in the milk of mothers with colicky infants. This finding is consistent with a study of bacterial lipopolysaccharide-mediated mastitis in mammary epithelial cells, which reported perturbations in both let-7a-5p and milk cytokines in response to bacterial antigens [55]. Transfer of these inflammatory components to the infant gut may contribute to an inflammatory state and produce the characteristic symptoms of infant colic (Figure 6). Indeed, strains of *Staphylococcus epidermidis* in human milk display marked similarity to those found in the stool of breastfeeding infants, suggesting the potential for mother–infant transfer [56]. Although not all strains of *Staphylococcus* are pathogenic, *Staphylococcus aureus*, in particular, has been associated with pathologic conditions in both the mammary glands and the infant gut [57,58].

Strengths of the current study include a prospective design, including a large number of breastfed infants with longitudinal outcome measures. The study also benefits from a rich collection of medical/demographic, social, and nutrition factors that allow us to control for a variety of important environmental variables while exploring the relationship between milk multi-omics and infant colic. However, there are several limitations that should be considered. The cross-sectional analysis of human milk samples prevents causational conclusions about the relationship between milk molecular profiles and infant colic symptoms. The majority of participants were white, and lacked socioeconomic risk factors (e.g., health insurance, poverty), which may limit generalizability. Given the differences in infant race/ethnicity that existed between colic and non-colic groups, it will be important to validate these findings in a more diverse cohort of infants. We acknowledge that the biologic framework developed in this study does not involve multi-omic analysis of infant stool. Therefore, strong conclusions about the impact of milk components on infant gut inflammation or dysbiosis cannot be drawn. Finally, in light of the relatively low incidence of infant colic (5–20%), which is replicated in the current cohort (28/182, 15%), larger case–control studies will be required to validate these findings, while taking into consideration additional factors such as exterogestation. 

## 5. Conclusions

The findings of this study support the idea that a complex network of microbial, protein, and miRNA factors within human breast milk may impact infant risk for colic development. These factors support prior hypotheses involving dysbiosis, intestinal epithelial permeability, and inflammation. If validated in larger case–control studies, they may serve as novel therapeutic targets for a medical condition that currently has limited treatment options. 

## Figures and Tables

**Figure 1 biomolecules-13-00559-f001:**
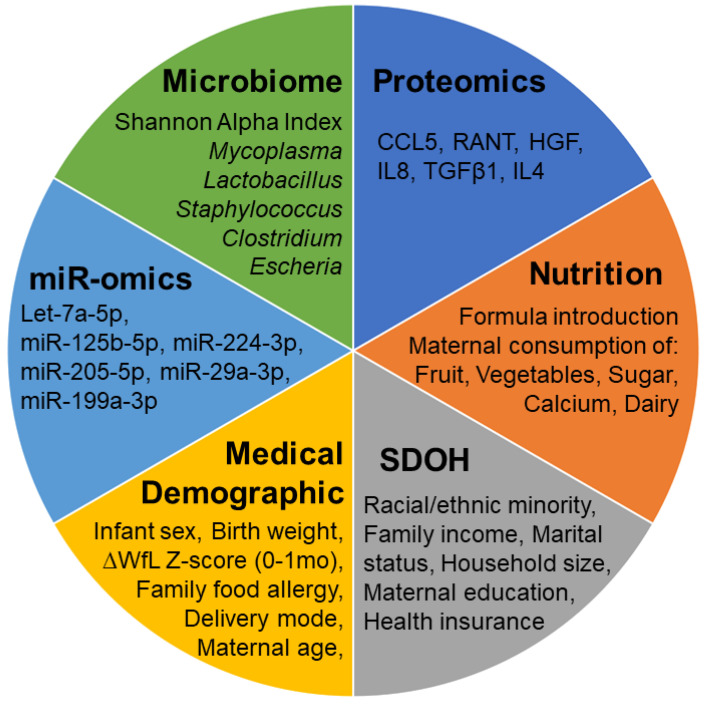
Conceptual framework for a multi-omics study of human milk factors involved in infant colic. The figure displays six categories of factors explored in this prospective cohort study of infant colic: Milk proteins (cytokines); Milk micro-ribonucleic acids (miR-omics); Milk microbiome; Medical/demographic traits; Social determinants of health (SDOH); and Nutritional components. Based on prior studies investigating these factors in nociception, gastrointestinal inflammation, and infant colic, we selected six features within each category for investigation.

**Figure 2 biomolecules-13-00559-f002:**
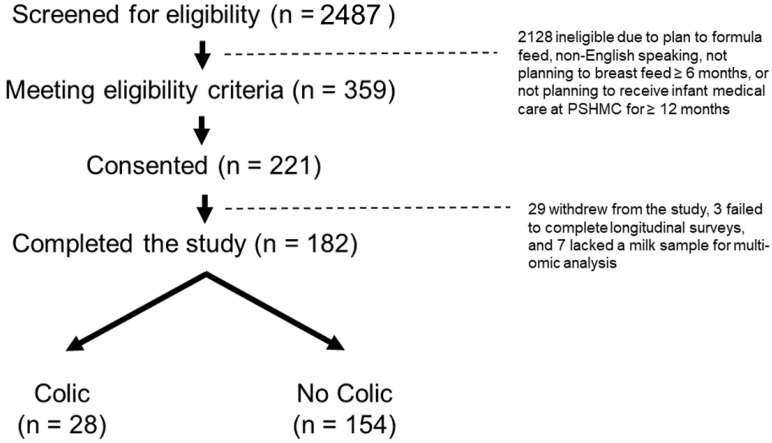
There were 2487 mother–infant dyads screened, 359 eligible dyads approached, and 221 dyads consented. There were 182 infants for whom human milk samples were available, surveys were completed, and colic status could be confirmed. Infant colic status was determined through a combination of clinician diagnosis (extracted from the electronic medical record), and maternal report on an abbreviated version of the Infant Colic Scale.

**Figure 3 biomolecules-13-00559-f003:**
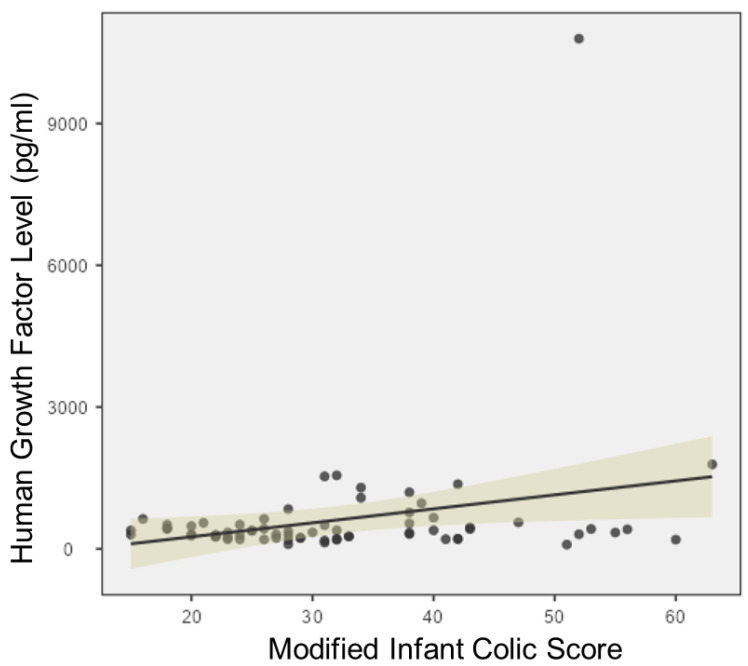
Levels of hepatocyte growth factor in human milk are directly associated with level of colic symptoms on the modified Infant Colic Scale. The scatter plot displays levels of hepatocyte growth factor (HGF) in human milk obtained at one month from mothers of infants with (*n* = 28) and without (*n* = 154) colic. HGF levels were directly correlated with the level of colic symptoms reported by mothers (*R* = 0.21, *p* = 0.025) on Spearman’s rank testing.

**Figure 4 biomolecules-13-00559-f004:**
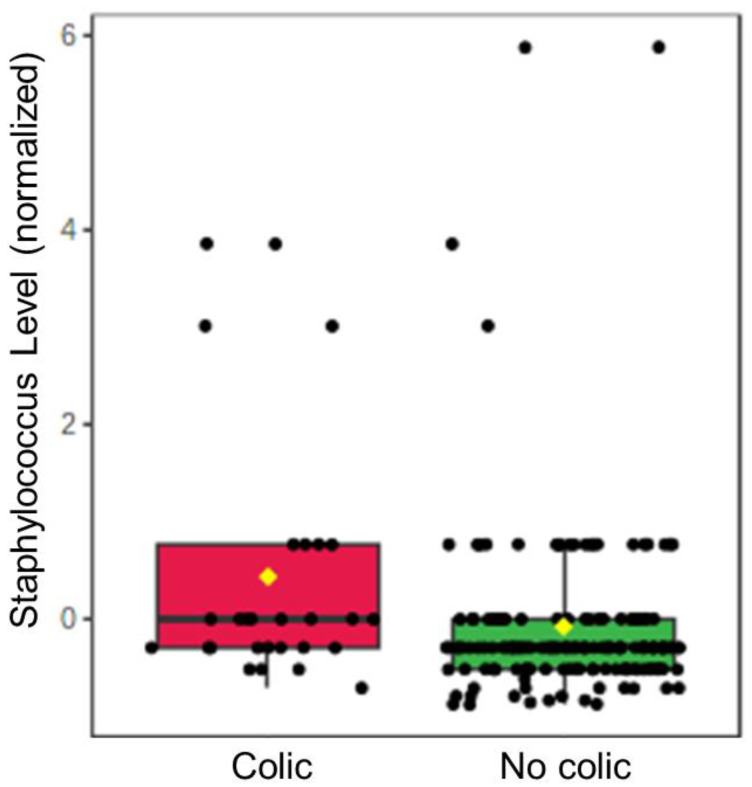
Levels of *Staphylococcus* in human milk are higher among mothers of infants with colic. The boxplot displays levels of *Staphylococcus* in human milk obtained at one month from mothers of infants with (red) and without colic (green). Quantile normalized levels of *Staphylococcus* transcripts were higher in samples from the colic group (*p* = 0.0076, adj. *p* = 0.038, *d* = 0.30) on Mann–Whitney testing. Mean (yellow diamond), median (black bar), and standard deviation are displayed.

**Figure 5 biomolecules-13-00559-f005:**
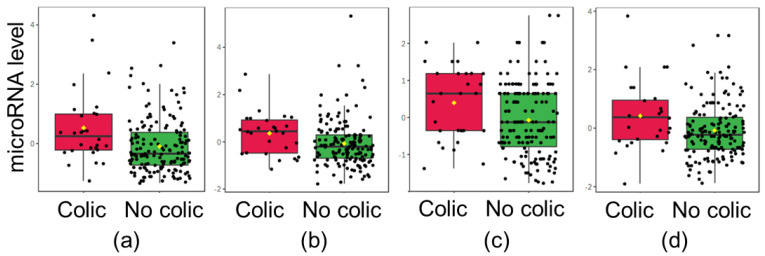
Levels of four miRNAs in human milk differ among mothers of infants with colic. Mann–Whitney testing revealed higher levels of (**a**) miR-224-3p (*p* = 0.0038, adj. *p* = 0.023, *d* = 0.33), (**b**) miR-125b-5p (*p* = 0.010, adj. *p* = 0.028, *d* = 0.29), (**c**) let-7a-5p (*p* = 0.018, adj. *p* = 0.028, *d* = 0.27), and (**d**) miR-205-5p (*p* = 0.019, adj. *p* = 0.029, *d* = 0.26) in the milk from mothers of infants with colic (red), compared to milk from mothers of infants without colic (green). Quantile normalized levels from RNA sequencing are displayed, with mean (yellow diamond), median (black bar), and standard deviation.

**Figure 6 biomolecules-13-00559-f006:**
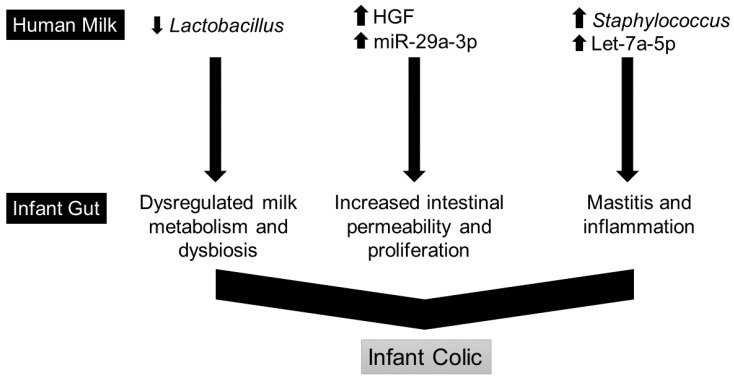
Multi-omic network implicated in infant colic. The findings of the current study support three common mechanisms that have previously been implicated in infant colic: dysbiosis, intestinal permeability, and inflammation. Reductions in milk *Lactobacillus* may drive dysbiosis, while perturbations in miR-29a-3p and hepatocyte growth factor (HGF) promote permeability. Finally, mastitis conditions involving *Staphylococcus aureus* may yield inflammatory milk components such as let-7a-5p, which can be transferred to the infant gut.

**Table 1 biomolecules-13-00559-t001:** Participant characteristics. Medical/demographic factors, social determinants of health, and nutrition factors are displayed for infants with colic and peers without colic.

Participant Characteristics	All (*n* = 182)	Infant colic (*n* = 28)	No colic (*n* = 154)
*Medical and demographic characteristics*			
Maternal age, years, mean (SD)	29 (4)	31 (4)	29 (4)
Female sex, *n* (%)	107 (58)	16 (57)	91 (59)
Vaginal delivery, *n* (%)	146 (80)	20 (71)	126 (81)
Birth weight, grams (IQR)	3357 (569)	3283 (454)	3370 (593)
∆WfL Z-score (0–1 months), mean (SD)	1.12 (1.5)	1.29 (1.0)	1.09 (2.1)
Family food allergy, *n* (%)	17 (9)	3 (10)	14 (9)
*Social determinants of health*			
Infant racial/ethnic minority, *n* (%)	43 (23)	11 (39) *	32 (20)
Family income < federal poverty level, *n* (%)	10 (5)	3 (11)	7 (4)
Married, *n* (%)	149 (81)	21 (75)	128 (83)
Persons in household, median (range)	4 (2–9)	4 (2–9)	4 (2–9)
Maternal college diploma, *n* (%)	131 (72)	21 (75)	110 (71)
Private health insurance, *n* (%)	157 (86)	24 (85)	133 (86)
*Nutrition factors*			
Formula introduced by 1 month, *n* (%)	57 (31)	13 (46)	44 (28)
Maternal vegetables, cups/day mean (IQR)	1.56 (0.4)	1.63 (0.6)	1.55 (0.4)
Maternal fruit, cups/day, mean (IQR)	1.16 (0.6)	1.26 (0.6)	1.14 (0.6)
Maternal sugar, teaspoon/day mean (IQR)	18.6 (6)	21.3 (8)	18.2 (6)
Maternal calcium, mg/day, mean (IQR)	1055 (232)	1036 (206)	1058 (233)
Maternal dairy, cups/day, mean (IQR)	1.89 (0.6)	1.85 (0.6)	1.90 (0.5)
*Human milk factors*			
Infant age at collection, days (SD)	39 (12)	38 (10)	39 (12)
Time of day at milk collection, hour (SD)	12 (2)	12 (3)	11 (2)

* Denotes *p* < 0.05 on chi-square test. Income data were only provided for 170 mother–infant dyads (27 colic, 143 non-colic). Abbreviations: weight-for-length (WfL), months.

## Data Availability

The RNA sequencing data presented in the study are deposited in the Gene Expression Omnibus repository, accession number GSE192543. GEO repository link: https://www.ncbi.nlm.nih.gov/geo/query/acc.cgi?acc=GSE192543; 1 January 2022.

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
