# Peer review of "The Association between Infant Colic and the Multi-Omic Composition of Human Milk"

_biomolecules, 2023, doi:10.3390/biom13030559_

Round 1

Reviewer 1 Report

The paper by Chandran et al., is aimed to analyze human milk components, including miRNA, proteins, and microbial members, that could be related with the presence of colic in infants. Their main results are that abundance of Lactobacillus and Staphylococcus, as well as higher levels of HGF and the miRNA let-7a-5p could be involved in the risk of colic. The manuscript is well written addresses a valid question and the results are novel; however, it would benefit from some revisions as requested below.

Introduction:

1.     It will be desirable to briefly include in the introduction the rationale for fusing on specific taxa in the comparison between colic vs non colic infants.

2.     Figure 2: The flow diagram lead to two groups referring to food allergy, should that be colic?

3.     Line 118: The authors state that infant colic status was determined through a combination of clinician diagnosis, could the authors extend on what were these criteria.

4.     Line 166: In the methods section it is described that the lipid fraction was used for RNA extraction, particularly related to the advantages of miRNA, is it possible that this could have influence the microbial profile observed in milk samples?

5.     Line 333-334: The authors state that the study identifies bacterial pathogens (i.e., staphylococcus), which are elevated in the milk of mothers with colicky infants. Although certain strains of Staphylococcus are pathogenic, this genus is part of the core milk microbiome, and it is suggested that aid in the oxygen consumption in the infant gut. How do the authors reconcile these findings?

6.     Line 327-330: The authors explain the higher levels of HGP observed in colicky infants represent a compensatory response of human milk composition to unique infant needs, that has been shown mainly in preterm infants. Since the infants included in the study were term babies, could the authors discuss the possible “needs” of these infants.

7.     Despite the prospective nature of the study, in the current analysis only one time point was evaluated, thus the authors should include, as a limitation of the study, that causality cannot be inferred.

Reviewer 2 Report

Thank you for submitting the manuscript "The association between infant colic and the multi-omic composition of human milk" to Biomolecules.

The manuscript reports results obtained from research that relates molecules present in breast milk with the occurrence of colic in infants. Indeed, the results are promising, and the research appears to have been conducted within satisfactory experimental and ethical standards. However, I do have a few issues to point out.

Line#41: italic

Line#110: How did the professionals exclude the possibility that the symptoms were linked to "exterogestation" and not the colic itself?

Line#125: Has research into the use of dietary supplements such as probiotics been carried out? It can influence the presence of microorganisms/molecules in breast milk and needs to be described in this work.

Line#152: point

Line#226: Based on the results of the work it is possible to exclude that the colic group is not being confused with the infant's temperament and/or symptoms of exterogestation.

Line#253: Is it possible to associate the high occurrence of Staphylococcus with foods present in the diet?
